# The Effect of Temperature on the Embryo Development of Cephalopod *Sepiella japonica* Suggests Crosstalk between Autophagy and Apoptosis

**DOI:** 10.3390/ijms242015365

**Published:** 2023-10-19

**Authors:** Yifan Liu, Long Chen, Fang Meng, Tao Zhang, Jun Luo, Shuang Chen, Huilai Shi, Bingjian Liu, Zhenming Lv

**Affiliations:** 1National Engineering Research Center for Marine Aquaculture, Zhejiang Ocean University, Zhoushan 316022, China; et999927@163.com (Y.L.); m18868010617@163.com (F.M.); 2Marine Science and Technical College, Zhejiang Ocean University, Zhoushan 316022, China; cl18969250500@163.com (L.C.); lj53296267@163.com (J.L.); 17861513027@163.com (S.C.); 3Zhejiang Province Key Lab of Mariculture and Enhancement, Marine Fisheries Research Institute of Zhejiang, Zhoushan 316021, China; zhangtao5729@163.com (T.Z.); shihuilai1980@163.com (H.S.)

**Keywords:** embryonic development, temperature, autophagy, apoptosis, *Sepiella japonica*

## Abstract

Temperature is a crucial environmental factor that affects embryonic development, particularly for marine organisms with long embryonic development periods. However, the sensitive period of embryonic development and the role of autophagy/apoptosis in temperature regulation in cephalopods remain unclear. In this study, we cultured embryos of *Sepiella japonica*, a typical species in the local area of the East China Sea, at different incubation temperatures (18 °C, 23 °C, and 28 °C) to investigate various developmental aspects, including morphological and histological characteristics, mortality rates, the duration of embryonic development, and expression patterns of autophagy-related genes (*LC3*, *BECN1*, *Inx4*) and apoptosis marker genes (*Cas3*, *p53*) at 25 developmental stages. Our findings indicate that embryos in the high-temperature (28 °C) group had significantly higher mortality and embryonic malformation rates than those in the low-temperature (18 °C) group. Furthermore, high temperature (28 °C) shortened the duration of embryonic development by 7 days compared to the optimal temperature (23 °C), while low temperature (18 °C) caused a delay of 9 days. Therefore, embryos of *S. japonica* were more intolerant to high temperatures (28 °C), emphasizing the critical importance of maintaining an appropriate incubation temperature (approximately 23 °C). Additionally, our study observed, for the first time, that the Early blastula, Blastopore closure, and Optic vesicle to Caudal end stages were the most sensitive stages. During these periods, abnormalities in the expression of autophagy-related and apoptosis-related genes were associated with higher rates of mortality and malformations, highlighting the strong correlation and potential interaction between autophagy and apoptosis in embryonic development under varying temperature conditions.

## 1. Introduction

As a crucial environmental factor, temperature significantly impacts the embryonic development of aquatic organisms. Previous research has highlighted the high sensitivity of embryonic development and sexual maturity of aquatic organisms to temperature [1]. Unlike fish, cephalopod embryos undergo a longer development process, typically lasting approximately 18–20 days for *Sepiella japonica* at a water temperature of 23–26 °C [2,3]. During this crucial period, temperature exerts a particularly influential role in embryonic development and incubation duration. Several studies on cephalopods have demonstrated an inverse relationship between the duration of embryonic development and temperature within a specific range [4,5]. Furthermore, temperature also significantly impacts the mortality or survival rate of cephalopod embryos. Presently, research on the relationship between embryo development and temperature in *S. japonica* primarily focuses on the impact of temperature on the developmental cycle and cumulative survival rate [2,6]. However, there remains a lack of investigation into the occurrence of embryonic malformation resulting from abnormal temperatures and the molecular-level regulatory mechanism of relevant genes and pathways.

Autophagy, a crucial and evolutionarily conserved process in eukaryotes, serves as a response to various stress stimuli. In this process, damaged proteins or organelles are encapsulated within autophagic vesicles with a bilayer membrane structure and transported to lysosomes (in animals) or vesicles (in yeast and plants) for degradation and recycling. This process plays a pivotal role in maintaining cellular homeostasis [7,8,9]. Under normal circumstances, autophagy is maintained at a certain baseline level, and it can be induced by various internal and external factors. Internal factors include broken senescent organelles and misfolded proteins, while external factors encompass nutrient deficiency, hypoxia, and temperature [10]. Although the impact of temperature on autophagy has been predominantly studied in cell lines and adult organisms [11,12,13], research in various plant and animal model organisms has indicated that high or low temperatures can activate autophagy to varying degrees [14,15,16,17].

Autophagy and apoptosis often coexist in biological processes such as embryonic development, tumorigenesis, and organism aging [18,19,20]. Apoptosis, known as type I programmed cell death, significantly influences organism development, cell renewal, and internal environment stability [21,22]. Studies in species including pufferfish *(Takifugu obscurus)*, white shrimp (*Litopenaeus vannamei*), and disk abalone (*Haliotis discus hannai*) found that excessively high or excessively low temperatures can cause cell apoptosis, resulting in abnormal physiological activities of the organisms [23,24,25]. Despite this, limited research exists on the exact relationship between autophagy/apoptosis and temperature during the embryo development process [12,13,26].

Innexin (Inx) is a highly conserved gap junction protein found exclusively in invertebrates [27,28]. It shares a similar membrane topology and quaternary structure with the vertebrate gap junction protein connexin (Cx). Gap junction intercellular communication (GJIC) studies have demonstrated its crucial role in the apoptosis regulation [29,30,31,32]. Additionally, the involvement of connexin in autophagy has been observed in various studies. For instance, in mouse hepatocytes, Cx43 participates directly in the initial stage of autophagosome formation [33]. In rats, the phosphorylation of Cx43 in astrocytes is positively correlated with autophagy intensity, and inhibiting this phosphorylation weakens autophagy occurrence [34].

Cephalopoda, a class of Mollusca comprising 901 known species [35], includes some cuttlefish, which are easily bred in artificial breeding [36]. Thus, they serve as suitable model species for studying the interaction of the changing environment on morphological and physiological aspects. However, the sensitivity of cephalopod embryos at different developmental stages to temperature has remained as mysterious as this group itself. *S. japonica*, commonly known as cuttlefish, holds important economic and medicinal value and is one of the “four major seafoods” in the East China Sea. It used to have a large yield, but now it is mentioned in the IUCN Red List of Threatened Species. Local scientists have invested a lot of effort in breaking through its artificial breeding technology and using it in resource rebuilding. To date, research on *S. japonica* has primarily focused on reproductive behavior, gonad development, and immune response [37,38,39,40]. Little is known about the specific effects of different temperatures on morphological characteristics, autophagy, and apoptosis during embryo development.

Building on the research background mentioned above, this study aims to establish a temperature gradient model (18 °C, 23 °C, and 28 °C) to investigate the effect of high and low temperatures on survival rate, tissue structure changes, developmental duration, and expression characteristics of genes related to autophagy (*LC3*/*BECN1*/*Inx4*) and apoptosis (*Cas3*/*p53*) in *S. japonica* embryos at different developmental stages. The objective is to gain initial insight into the interaction between autophagy and apoptosis during embryonic development under varying temperatures. The findings lay a groundwork for understanding how environmental factors like temperature influence embryo development of this mysterious cephalopod. Additionally, this research contributes to optimizing artificial nursery conditions, improving the survival rate of young adults, and offering valuable insights for the successful recovery of this precious species. The study holds significant scientific and industrial application value.

## 2. Results

### 2.1. Mortality Rate and Cumulative Survival Rate

The mortality rate and cumulative survival curve at different embryonic development stages under different temperatures are shown in Figure 1. In the Ctrl group, sporadic deaths of the embryos occurred at the Early blastula and Cuttlebone stages, with an overall mortality rate remaining below 4.5% before hatching (Figure 1A). In the low-temperature group (18 °C), significant mortality first occurred at the Multicellular stage, and then mainly occurred at the Anlage, Heart beating, and Cuttlebone stages. The mortality rate remained low until 4 DPH (day post hatching), with the highest death rate at 5 DPH (Figure 1B). In the high-temperature group (28 °C), significant mortality first occurred at the Anlage stage, then at the Heart beating and Cuttlebone stages, reaching its peak at 3 DPH, followed by a gradual decrease (Figure 1C).

The cumulative survival rates (Figure 1D) in the Ctrl group remained above 98.55 ± 2.96% before hatching and above 64.73 ± 3.24% until the end of the experiment (5 DPH). In the low-temperature group (18 °C), the cumulative survival rate started to decline slightly at the Early blastula stage and decreased significantly at the Chromatophore and Caudal end stages before hatching. The cumulative survival rate remained above 82.42 ± 4.12% until hatching and dropped to 76.32 ± 3.82% at 5 DPH. In the high-temperature group (28 °C), the cumulative survival rate began to decrease significantly at the Chromatophore stage and decreased again at the Caudal end stage. The cumulative survival rate decreased to 83.23 ± 4.99% before hatching. It decreased significantly at 4 DPH due to the mass death at 3 DPH, resulting in a cumulative survival rate of 42.90 ± 8.58% at the end of 5 DPH. Significant decreases in cumulative survival rate usually occurred at the stage following the high mortality period.

### 2.2. Effects of Different Temperatures on the Embryonic Development Duration

Different temperature treatments also caused changes in the duration of certain developmental stages (Figure 2). Embryos exposed to low temperature (18 °C) exhibited a delay starting at the four-cell stage, with an average delay of 0.5 days in all subsequent periods. This resulted in a total developmental length delay of approximately 9 days compared to the Ctrl group. The delay was more significant at the Early blastula and Heart beating stages (*p <* 0.05). Conversely, embryos cultured at high temperature (28 °C) showed developmental advancement at the four-cell stage. Each subsequent developmental period occurred, on average, 0.39 days earlier than the Ctrl group, resulting in a total advancement of 7 days in the overall embryonic development duration. The most notable advancement was observed at the Blastopore closure stage. The largest differences between the high and low temperature groups were observed at the Chromatophore stage, followed by the Cuttlebone, Caudal end, and Anlage stages.

### 2.3. Embryonic Malformation Caused by Low and High Temperatures

To investigate the impact of low and high temperatures on embryo morphology, we compared their morphological and histological changes occurring at the most sensitive stages, as depicted in Figure 3. In the Ctrl group, no embryonic malformations were observed. However, in the low-temperature group (18 °C), the rate of embryonic malformation was 12.97 ± 0.73%. Specifically, shortened arms and disordered external organ development were observed at the Optic vesicle stage and Heart beating stage, respectively. On the other hand, the high-temperature group (28 °C) exhibited an even higher embryonic malformation rate of 20.04 ± 0.57%. At the Chromatophore and Caudal end stages, swelling of the visceral mass, exposed gills, and ectropion mantle were observed.

Further analysis of the histological changes at the Caudal end stage revealed significant malformations (Figure 3B). Compared to the Ctrl group, embryos in the 18 °C group exhibited swelling of the visceral mass and a tendency towards ectropion mantle, which caused the changes in embryonic tension, resulting in abnormal distortions of the body, brain, and arms, exhibiting abnormal curvature resembling a “V” shape. Additionally, in the high-temperature group (28 °C), the morphology of the mantle was also distinctly different from that of the Ctrl group, along with a tendency towards more serious ectropion.

### 2.4. Expression of Autophagy Marker Genes LC3/BECN1 of the Embryos

To investigate the impact of low and high temperatures on autophagy levels in embryos, we conducted a systematic study using the common autophagy marker genes *LC3* and *BECN1*. The results are presented in Figure 4. In the Ctrl group, the expression of *LC3* fluctuated, with high levels observed at the Late blastula, Late gastrula, and Heart beating stages, and the highest expression was observed at the Heart beating stage and 4 DPH. The expression patterns of *LC3* in the low-temperature (18 °C) and high-temperature (28 °C) groups were similar to the those in Ctrl group before hatching, with fluctuating expressions and a peak at the Heart beating stage. In the 18 °C group, the second highest expression was observed at the Chromatophore stage. Furthermore, the expressions of *LC3* at these two stages (Heart beating and Chromatophore) were significantly higher than those in the Ctrl group. In the high-temperature group (28 °C), the overall expression level of *LC3* was also much higher than in the Ctrl group. However, the expression pattern in the 28 °C group was more similar to that of the Ctrl group (early fluctuations, high expression at the Heart beating stage and 4 DPH), and the two periods of highest expression (before hatching) were consistent with the 18 °C group (highest expressions observed at the Heart beating and Chromatophore stages). Additionally, a significant increase in *LC3* expression at the Chromatophore stage was observed in both temperature treatment groups.

However, both low- and high-temperature treatments significantly altered the expression pattern of *BECN1* (Figure 4B). In the Ctrl group, *BECN1* remained at a low level until reaching the Optic vesicle stage (highest expression) and Heart beating stage (second-highest expression). However, both low and high temperatures caused a significant increase in *BECN1* expression during the early developmental stages, from the Zygote to the Multicellular stage, with similar “M”-shaped curves. In both the 18 °C and 28 °C groups, the highest expression occurred at the two-cell stage, followed by a significant decrease at the four-cell stage. After another increase around the 8-cell to 16-cell stages, *BECN1* expression decreased again at the Early blastula stage. From then on, it maintained a lower expression level until the end of the experiment. Compared to the Ctrl group, temperature treatments resulted in increased *BECN1* expression during the early stages and a significant decrease at the Optic vesicle and Heart beating stages.

### 2.5. Expression of Autophagy-Related Gene Inx4 of the Embryos

We also examined the expression changes in the potential autophagy-related gene *Inx4* under different temperatures, as shown in Figure 5. In the Ctrl group, *Inx4* reached its peak at the Heart beating stage, with relatively low expression levels during other periods. However, both low- and high-temperature treatments significantly increased the expression of *Inx4*, especially at the Heart beating stage in the high-temperature group (28 °C), which was approximately 45 times higher than in the Ctrl group. Additionally, there were other stages at which *Inx4* expression significantly increased, including the Chromatophore stage in the 18 °C group, and the four-cell, External organ, and Chromatophore stages in the 28 °C group. The periods of high *Inx4* expression were directly influenced by the temperature treatment, with high temperature having a significant induction effect on *Inx4* expression.

### 2.6. Expression of Apoptosis Marker Genes Cas3/p53 of the Embryos

To further investigate the effect of temperature on embryo apoptosis, we analyzed the expressions of the apoptosis-related genes *Cas3* and *p53* in the control (23 °C), low-temperature (18 °C), and high-temperature (28 °C) groups. As shown in Figure 6A, higher expressions of *Cas3* were mainly observed from the blastula to the Chromatophore stages. In the Ctrl group, *Cas3* expression first increased and then decreased, with the highest expression at the Late gastrula stage. In the 18 °C group, *Cas3* expression showed similar trends, with the highest expression point shifting to the Anlage stage, and this relatively high level of *Cas3* expression was consistently maintained until the Chromatophore stage. In the 28 °C group, the highest *Cas3* expression was observed at the Optic vesicle stage.

The results for *p53* in all groups, especially the temperature treatment groups, showed a similar trend of variation (Figure 6B). In the control group, *p53* expression was higher at the Zygote, Late blastula, and Anlage stages. In both the low-temperature (18 °C) and high-temperature (28 °C) groups, *p53* expression was higher at the four-cell stage, blastula, and Anlage stages.

### 2.7. Heat Map of the Gene Expression Elevated Folds

To visualize the overall changes in gene expression under different temperatures, we analyzed the fold changes in expression of all autophagy and apoptosis-related genes. As shown in Figure 7, the periods of increased *LC3* expression in the low- and high-temperature groups mainly occurred from the Zygote to the Early blastula stage and from the Anlage to the Cuttlebone stage. The expression patterns of *BECN1* in the low- and high-temperature groups were also similar. In contrast to the low-temperature group, the expression of *Inx4* in the high-temperature group showed significant differences compared to the Ctrl group at almost every stage. Additionally, the expression patterns of the apoptosis marker genes (*Cas3*/*p53*) in the embryos of the low-temperature group and high-temperature group were similar, with no significant differences in other periods except for a few stages. Furthermore, compared to the Ctrl group, the expressions of autophagy-related genes (*LC3*/*BECN1*/*Inx4*) and apoptosis marker genes (*Cas3*/*p53*) all increased to varying degrees at the Chromatophore stage, regardless of whether it was the low- or high-temperature group. It is worth noting that the increased expression of *Inx4* in the 28 °C group was significantly higher than in the 18 °C group.

### 2.8. Apoptotic Staining (TUNEL) of Embryos at Sensitive Stages

Based on the previous findings, we selected embryos at the Caudal end stage, known for typical periods of embryonic malformations periods, from the control and temperature treatment groups. Using TUNEL staining, we further investigated the apoptosis levels of the embryos. The enlarged images were subsequently divided into three sections for presentation purposes: body, head, and arms (Figure 8A–C). DAPI staining visualizes nuclei (blue), TUNEL staining indicates apoptosis (red), and strong apoptosis signals are marked by white arrows.

Compared to the Ctrl group, the low-temperature group (18 °C) showed concentrated apoptotic signals mainly in the arms, while the high-temperature group (28 °C) exhibited signals mainly in the head and arms. Overall, the 28 °C group displayed more and stronger signals than the 18 °C group, highlighting the temperature-dependent impact on apoptosis during embryonic development.

## 3. Discussion

Among the environmental factors influenced by global change, temperature is, by far, the most documented as it concerns the effects on developmental plasticity [41]. Understanding favorable or optimal environmental conditions is crucial for the artificial propagation of any species [42]. Water temperature is a key abiotic factor for the survival of aquatic organisms and plays a critical role in successful hatchery production [43]. Furthermore, changing oceanic water temperatures during early life stages is crucial for species recruitment and survival. Especially drastic temperature changes have obvious effects on the development of embryos and larvae, mainly reflected in the survival rate. For instance, in the study of *Acipenser ruthenus* L, it was found that cold shock led to a significant decline in the survival rate of embryos [44]. What is more, previous studies showed that high temperatures lead to higher mortality rates than low temperatures in the Asian yellow pond turtle (*Mauremys mutica*) [45], tehuelche octopus (*Octopus tehuelchus*) [46], and zebrafish [47]. In the present study, compared to the Ctrl group, temperature treatments resulted in increased embryonic mortality, especially in the high-temperature group. We speculate that this is due to the slower metabolism and weaker stress response of embryos in the low-temperature group [48], resulting in a higher survival rate compared to the high-temperature group. Many embryonic and adult studies have found that the cellular damage caused by high temperatures is much greater than that caused by low temperatures, which is usually irreversible [49,50,51].

It is worth mentioning that in both the low- and high-temperature groups, periods of higher embryonic mortality mainly occurred during Heart beating and Cuttlebone stages. This suggests that these two periods are likely to be temperature-sensitive. However, in the temperature treatment groups, the embryonic development periods in which the cumulative survival rate significantly decreased occurred at the Chromatophore and Caudal end stages, slightly lagging behind the periods of high instantaneous mortality. Typically, significant decreases in cumulative survival rate occur in the stage following the high mortality period. Therefore, the lag phenomenon is caused by the accumulation of death.

The plasticity in the hatching time of cephalopods can provide an advantage for their survival under heterogeneous environmental conditions [52]. Hatching can be delayed if the embryonic developmental temperature is close to the lower temperature limit to which the species is adapted [53,54]. In contrast, within a certain temperature range, an increase in temperature shortens the duration of embryonic development [55]. For model animals such as zebrafish higher temperatures above 28.5 °C have been shown to accelerate zebrafish embryonic development [56]. For mollusca, higher water temperature increased the metabolism of the embryos and consequently accelerating development [57]. Especially in cephalopods, according to Uriarte (2012) [58] and Nande (2018) [59], a shorter embryonic development and hatching period due to an increase in temperature allows earlier hatching of octopuses, as well as faster growth and reaching of reproductive age. Embryos of *Amphioctopus fangsiao* hatched successfully at 40, 30, and 24 days post fertilization (dpf) at temperatures of 18 °C, 21 °C, and 24 °C, respectively [4]. Similar results were observed in this study: compared to the Ctrl group, the overall development time of embryos in the low-temperature group was delayed by approximately 9 days, with significant delays occurring at the Early blastula and Heart beating stages. Our results indicate that the overall development time of embryos in the high-temperature group was shortened by 7 days compared to the Ctrl group. This inverse relationship between temperature and incubation period is consistent with other species. Additionally, the first significant advancement in embryonic development of *S. japonica* occurred at the stage of Blastopore closure in the high-temperature group, while the first significant delay occurred at the stage of Early blastula in the low-temperature group. However, the most notable differences in the duration of embryonic development between the high- and low-temperature groups were observed specifically at the Chromatophore stage. Therefore, these stages, including Blastopore closure, Early blastula, and Chromatophore, may be abnormally sensitive to temperature in the embryonic development of *S. japonica*.

Abnormal temperatures not only impact the cumulative survival rate of embryos, but also lead to the occurrence of malformations. It has been extensively documented in fishes that the temperature during egg incubation, as well as the larval and juvenile periods, is a critical environmental factor that influences malformation rates [60,61]. Incubating zebrafish embryos at temperatures of 32.5 °C and above from 2.5 until 96 h post fertilization caused malformations occurring as early as 24 h post fertilization [56]. However, for mollusca, excessively high or excessively low incubation temperature causes a certain increase in the possibility of embryonic development malformation. Lower rates of deformities were observed during the embryonic stages of *C. seguenzae* at 23 °C, whereas development at 17 °C and 29 °C led to high rates of deformities or total mortality [62]. In *Coelomactra antiquata*, fertilized eggs developed to the larval stage after 20 h at a water temperature of 15.5 ± 0.5 °C but ceased to develop, while at a temperature of 28 ± 0.5 °C, the fertilized eggs became deformed [63]. Similarly, in *A. fangsiao*, where 25.9% of eggs incubated at a lower temperature (24 °C), the eggs failed to undergo inversion from the animal pole to the vegetal pole [4]. In the present study, malformations caused by low temperature (18 °C) primarily occurred during the stages of Optic vesicle and Heart beating, resulting in shortened arms and disordered external organ development. Conversely, malformations caused by high temperature (28 °C) mainly occurred during the stages of Chromatophore and Caudal end, characterized by swelling of the visceral mass, exposed gills, and ectropion mantle. Comparing the malformation characteristics between the two, it was observed that *S. japonica* embryos were more intolerant to high temperature (28 °C). Our experimental results demonstrate that both high-temperature (28 °C) and low-temperature (18 °C) incubation conditions induce malformations in *S. japonica* embryos, with high temperature displaying a more severe effect than low temperature.

Autophagy has been shown to be a survival mechanism under different stress conditions [64,65]. Furthermore, autophagy is an important response mechanism to temperature changes [6,66]. Current studies on the effect of temperature on the level of autophagy mainly focus on model organisms. In a study on the effects of high temperature on the growth of *Arabidopsis thaliana*, it was found that autophagy plays a crucial role in reducing the effects of high temperature on the growth of *A. thaliana* and maintaining specific pollen development [14]. Autophagy is also a major mechanism underlying heart remodeling in response to cold exposure and its subsequent reversion after deacclimation in mice [17].

In the present study, we found that the expression changes of *LC3* after high/low-temperature treatment tended to be generally consistent. The peak expression occurred during Heart beating and Chromatophore stages, indicating a high level of autophagy, which corresponded to the large number of malformations and high mortality observed at these two stages. The expression of *BECN1* showed a similar pattern of change after high/low temperature treatment, with a significant increase at the stages of 2-cell and 8–16-cell and a significant decrease at the stages of Optic vesicle and Heart beating, corresponding to the large number of malformations during these stages. This suggests the important role of *BECN1* in embryonic development. Similar results have been reported in mouse embryos [19], where mice lacking *BECN1* died at an early embryonic stage, and embryos with low expression did not survive beyond 8.5 days [67,68]. Based on the changes in the expression of *LC3*/*BECN1*, prolonged high/low-temperature stress significantly inhibited autophagic activity in embryos, resulting in massive embryonic malformation and death at sensitive stages, especially during Heart beating and Chromatophore stages.

In the Ctrl group, the expression of *Inx4* was highest at Heart beating stage. After high/low-temperature treatment, *Inx4* remained at a high level at the stage of Heart beating and significantly increased during Chromatophore stage. This corresponded to the significant number of embryonic malformations and high mortality observed around the stages of Heart beating and Cuttlebone. Notably, we found that after high/low-temperature treatment, there was a strong correlation between a significant increase in the expression of *Inx4* and high mortality. *Innexin*, as a unique gap junction protein gene in invertebrates, plays an irreplaceable role in embryonic development [69,70], immune response [71], and apoptosis [72,73]. Therefore, we speculate that the high expression of *Inx4* during sensitive stages is associated with embryonic malformations and high mortality, and may also be related to the apoptotic pathway. However, the direct regulatory relationship still needs to be elucidated through systematic validation.

There is a complex interactive regulatory relationship between apoptosis and autophagy [74]. The interaction between the two can achieve a dynamic balance to certain extent, which maintains the basic physiological functions of cells and reduces the damage to the body under stress [75]. Autophagy typically prevents the induction of apoptosis, while caspase activation associated with apoptosis inhibits the autophagy process [76,77]. In Nile tilapia (*Oreochromis niloticus*), the combination of apoptosis and autophagy modulate follicular atresia under heat stress [78]. Research on the tunicate (*Ciona intestinalis*) has highlighted the significant role of autophagy during the late phases of development in lecithotrophic organisms and has also helped identify the coexistence of autophagy and apoptosis in cells [79]. Similar results have been reported in mollusks, where cold and heat stress increased the activity of Cas8 in the gill tissues and blood cells of *Mytilus coruscus* and *Mytilus galloprovincialis*, leading to the activation of the apoptotic pathway [80]. In this study, the expression of *Cas3* was higher from blastula to Chromatophore stages in the Ctrl and high/low temperature groups. Compared to the Ctrl group, the expression of *Cas3* significantly increased in the low-temperature (18 °C) group from Anlage to Chromatophore stages, and in the high-temperature (28 °C) group at the Optic vesicle stage. Similarly, the expressions of *LC3* in the corresponding temperature treatment groups significantly increased compared to the Ctrl group. These findings indicate that the apoptotic pathway was significantly activated and the autophagic pathway was inhibited, corresponding to the increased mortality during the same periods.

Previous research found that high-temperature exposure up-regulated the activities of *p53*, *Bax*, *caspase 9*, and *caspase 3* in pufferfish (*T. obscurus*), and the *p53*-Bax pathway and the caspase-dependent apoptotic pathway were involved in high temperature stress-induced apoptosis in pufferfish blood cells [23]. Loss of the apoptotic factor Bruce in mouse fibroblast cells also caused a significant elevation in the expression of *p53* and activated *Pidd/cas2* and *Bax/Bak*, which induced the activation of *Cas3* and resulted in embryonic death [81]. In this study, the expression trends of *p53* in the control (23 °C), low (18 °C)-, and high (28 °C)-temperature groups were overall similar. However, compared to the Ctrl group, significantly higher expression mainly occurred at the four-cell, Early blastula, and Chromatophore stages in the high- and low-temperature groups. It is speculated that low and high temperatures activate the expression of *p53*, which in turn activates the proapoptotic genes Bax/Bak and triggers apoptosis. Therefore, it is speculated that either excessively high or excessively low incubation temperature causes abnormal activation of apoptosis-related signaling pathways, such as the *BECN1*-*Bcl2* pathway and the *p53* signaling pathway, which in turn activates the proapoptotic genes *Bax/Bak* and induces the activation of *Cas3*, triggering apoptosis. However, further verification is needed to understand the specific regulatory mechanism of this pathway.

In this study, TUNEL apoptosis staining results also showed that the apoptotic signal was stronger in the high-temperature group (28 °C) than in the low-temperature group (18 °C), indicating that *S. japonica* embryos were more sensitive and intolerant to excessive temperatures. Comprehensive analysis reveals that the phenomenon of abnormal elevation of gene expression, embryonic malformation, and death caused by high temperature is more severe than that caused by low temperature, suggesting that during embryo culture, high temperature should be treated more cautiously.

From the above research results, Early blastula, Blastopore closure, and stages from Optic vesicle to Caudal end are more sensitive periods during the *S. japonica* embryonic development. Considering the protection of germplasm resources and the supply of larval food, our findings advise fishery managers to pay more attention to the temperature when regulating the hatching time to meet the needs of protecting seedling proliferation and maximizing cost effectiveness associated with the incubation period. Meanwhile, during the embryonic development period which is more sensitive, fishery managers need to pay special attention to the culture conditions, which is of great importance in improving the hatching success rate. In addition, these sensitive periods and detection indicators can be used as a basis for screening embryo quality in the future, thereby achieving real-time monitoring and improving the *S. japonica* embryo culture.

## 4. Materials and Methods

### 4.1. Embryo Culture and Sampling

In April 2022, *S. japonica* adults (50 males and 25 females) were maintained and observed in an 80 m^3^ cement breeding pond (a semi-closed seawater system with one inlet and one outlet, 8 m × 5 m × 2 m) from a local cuttlefish breeding center (Xixuan Island, Zhejiang Province, China, 29°53′ N, 122°18′ E). Then, a 50% seawater was changed and the bottom was cleaned with automatic cleaning bottom suction machine twice a day. The broodstock were fed with shrimp twice a day. The seawater conditions were maintained as follows: temperature 19 ± 1 °C, salinity 27‰, dissolved oxygen level > 5 mg L^−1^, pH 6.8–7.2, and ammonia nitrogen level < 0.2 mg L^−1^. The photoperiod was 12 h light/12 h dark. The spawning nest (30 cm × 50 cm, with 18 # iron wire as the frame and sewn with 1 cm × 1 cm nylon mesh) was hung in the spawning pool for *S. japonica* to lay eggs. Upon spawning, around 4000 eggs (during the collection of fertilized eggs, each female laid an average of 160 eggs) were removed using sterilized forceps and suspended by a nylon rope in 25 L seawater tanks. Then, they were immediately transported to the laboratory of the National Marine Facilities Aquaculture Engineering Technology Research Center. Temporary culture conditions were as follows: tank size 40 cm × 22 cm × 28 cm, salinity 27‰, pH ranging from 6.8 to 7.2, ammonia nitrogen level < 0.2 mg L^−1^, continuous aeration, and 50% seawater renewal per day to maintain a dissolved oxygen concentration of more than 5 mg L^−1^. The photoperiod was 12 h light/12 h dark. Fresh seawater was pre-heated or pre-cooled to maintain stable temperatures in the different treatment groups: control group (Ctrl) at 23 °C, low-temperature group at 18 °C, and high-temperature group at 28 °C. Each group had three parallel replicates, with each replicate including approximately 450 fertilized eggs.

The hatching of fertilized eggs was regularly observed and recorded based on the developmental stages of the embryos. Referring to the embryonic stages as Naef (1928) [82] and Jiang (2020) [4], a total of 25 specific sampling periods were established, including Zygote, 2-cell, 4-cell, 8-cell, 16-cell, Multicellular, Early blastula, Late blastula, Early gastrula, Late gastrula, Blastopore closure, Anlage, External organs, Optic vesicle, Heart beating, Chromatophore, Cuttlebone, Caudal end, Veliger, Pre-hatching, 1 DPH, 2 DPH, 3 DPH, 4 DPH, and 5 DPH. The number of dead and malformed embryos, as well as the duration of each developmental stage, were recorded. At each stage, 12 embryos were randomly sampled from each replicate group. The embryos were stored at −8 °C for RNA analysis and placed in 4% paraformaldehyde (PFA) (Solarbio, Shanghai, China) for histological study.

All experiments were conducted in accordance with the protocols and guidelines approved by the Ethics Committee of Zhejiang Ocean University and the Academy of Experimental Animal Center of Zhejiang Ocean University.

### 4.2. Embryonic Development Observation and Histological Sections

Developmental features were observed under a stereoscopic microscope (Leica S9i, Wetzlar, HE, Germany). Before taking pictures, the outer two layers of egg membranes were gently peeled off, leaving three layers of egg membranes. Embryos without outer membranes were fixed in 4% PFA for 24 h, transferred to 50% Formamide deionized, and then stored at −20 °C.

For hematoxylin and eosin (HE) staining, the samples were frozen and sectioned using a frozen microtome (Leica CM3050s, Wetzlar, HE, Germany), and stored at −20 °C. The frozen sections were refixed in a 4% PFA for 10 min, rinsed twice with distilled water for 2 min each. The sections were stained with a hematoxylin and eosin (HE) staining solution (Phygene, Fujian, China): hematoxylin was applied for 5 min, rinsed with tap water until the samples turned blue; the sections were then placed in a 1% HCl solution for 5 s, rinsed with tap water again, followed by immersion in distilled water for 2 min, 50% ethanol for 2 min, 70% ethanol for 2 min, 80% ethanol for 2 min, eosin staining solution for 2 min, 95% ethanol for 2 min, 100% ethanol for 2 min, 50% xylene and 50% ethanol mixture for 2 min, and finally xylene for 5 min, twice. The sections were sealed with neutral resin and photographed under a microscope (Nikon NI-U, Tokyo, Japan).

### 4.3. RNA Extraction and cDNA Synthesis

Total RNA from tissues was isolated using a Trizol reagent (Takara, Kyodo, Japan) following the methods described in previous studies [83]. Before hatching, 6–8 eggs were taken as one sample pool at each developmental stage, and there were three parallel repetitive sample pools for each temperature treatment group. After hatching (1–5 DPH), 5–6 individuals were taken as a sample pool at each developmental stage, and there were also three parallel repetitive sample pools for each temperature group. The quality, purity, and integrity of the RNAs were assessed using UV spectroscopy (A260/A280) on Nanodrop 2000 (Thermo Fisher Scientific, Waltham, MA, USA) and agarose gel electrophoresis. The mRNA was reverse transcribed into first-strand cDNA using the PrimeScriptTM RT reagent Kit with gDNA Eraser (Perfect Real Time) (Takara, Kyodo, Japan). The first-strand cDNA was used as the template.

### 4.4. Real-Time Fluorescence Quantitative PCR Assay

Fluorescence quantification was performed using the SYBR Green dye method, and the internal reference genes *β-actin* (JN564496.1) and *GAPDH* [84] were selected to minimize bias at each stage. RT-qPCR experiments were performed in a final volume of 10 µL containing 0.6 µL of cDNA, 0.4 µL of each 10 mM gene-specific primer (Appendix A), and 5 µL of TB Green^®^ Premix Ex TaqTM II (Tli RNaseH Plus) (Takara, Kyodo, Japan) on a CFX Connect Real-time PCR amplifier (Bio-Rad, Richmond, VA, USA). The protocol was 95 °C (30 s) for heat denaturing; then, 40 cycles of 95 °C (5 s), 58 °C (30 s), 72 °C (20 s), and additional 72 °C (2 min). The amplification efficiency of each pair of primers was determined by a gradient dilution of template. The reactions were performed with three individual repeats and three independent experiments in each developmental stage. PCR specificity was assessed in the terms of a melting curve. The levels of *LC3*/*BECN1*/*Inx4*/*Cas3*/*p53* mRNA were analyzed using the threshold and Ct (threshold cycle) values according to the 2^−ΔΔCt^ method [85].

### 4.5. Apoptotic Staining (TUNEL)

Apoptotic staining was performed using the One-step TUNEL Cell Apoptosis Detection Kit (KeyGEN Biotech, Nanjing, China) with the red TRITC labeled fluorescence detection method, following the manufacturer’s instructions. The stained samples were observed under a fluorescence microscope (Leica DMI8, Wetzlar, Germany).

### 4.6. Data Statistics

Only symptoms with a frequency of more than 75% were considered as meaningful typical deformity symptoms for recording and comparison. Data statistics were performed using statistical software SPSS 22.0 (SPSS Inc., Chicago, IL, USA). Data that did not conform to a normal distribution were transformed by a natural logarithm before further analysis. Significance testing was conducted using one-way ANOVA (Duncan’s multiple range tests). A *p*-value of less than 0.05 was considered statistically significant (*p <* 0.05). All data were presented as means ± standard deviation (SD).

## 5. Conclusions

In summary, by analysis, the study results of morphology, histology, and expression changes in autophagy marker genes (*LC3*/*BECN1*), autophagy-related gene (*Inx4*), and apoptosis-related genes (*Cas3*/*p53*) during the embryonic development of *S. japonica* at different temperatures revealed that the most sensitive developmental stages were the Early blastula, Blastopore closure, and stages from Optic vesicle to Caudal end. *S. japonica* embryos exhibited greater sensitivity and intolerance to excessive temperature, with specific manifestations such as significantly higher embryo mortality in the high-temperature (28 °C) group compared to the low-temperature (18 °C) group. Moreover, embryonic malformations caused by high temperature (28 °C) were more severe, characterized by swelling of the visceral mass, exposed gills, and ectropion mantle. Furthermore, an increase in temperature (28 °C) led to a shorter embryonic development duration, while a decrease in temperature (18 °C) had the opposite effect. Therefore, our study demonstrated for the first time that maintaining the appropriate incubation temperature (around 23 °C) during these sensitive stages (Early blastula, Blastopore closure, and stages from Optic vesicle to Caudal end) is crucial for survival and normal development of *S. japonica* embryos. High temperature (28 °C) caused by heat waves or cultivation conditions is more harmful to embryos than low temperature (18 °C). Additionally, we observed that the sensitive stages of gene changes related to autophagy/apoptosis roughly corresponded to the sensitive stages of malformation and high mortality, indicating a strong correlation and potential crosstalk between autophagy and apoptosis in embryonic response to temperature, which also provides intriguing insights for future research.

## Figures and Tables

**Figure 1 ijms-24-15365-f001:**
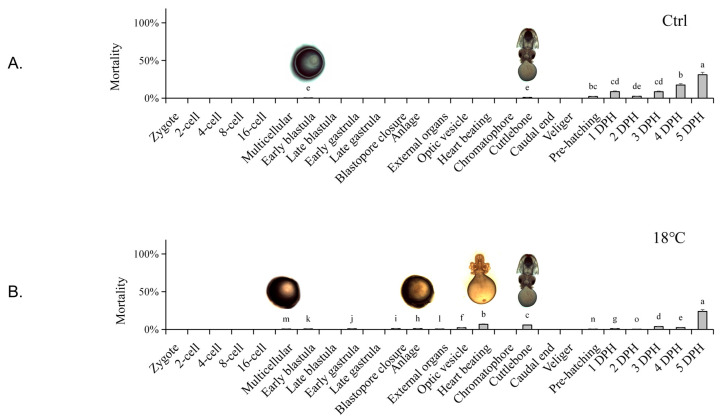
Mortality rate and cumulative survival curve of *S. japonica* embryos at different developmental stages under different temperatures. (**A**) Ctrl group (23 °C); (**B**) Low-temperature group (18 °C); (**C**) High-temperature group (28 °C); (**D**) Cumulative survival curve of all temperature groups. Embryo pictures represent typical periods. The values are means ± SD (n = 12). “*” denotes significant differences between different groups (*p* < 0.05). Different letters denote significant differences between different groups (*p* < 0.05).

**Figure 2 ijms-24-15365-f002:**
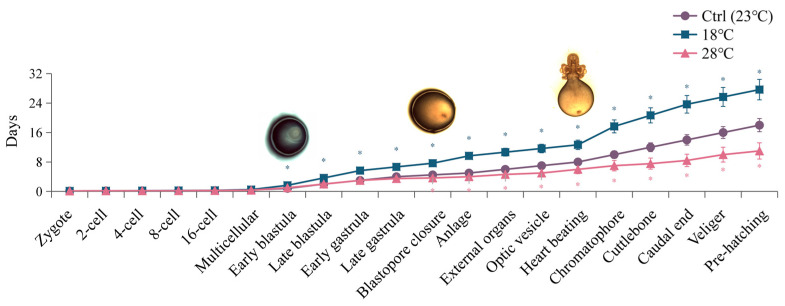
Embryonic development duration of *S. japonica* embryos under different temperatures. The values are means ± SD (n = 12). Embryo pictures represent typical periods. “*” denotes significant difference between different groups (*p* < 0.05).

**Figure 3 ijms-24-15365-f003:**
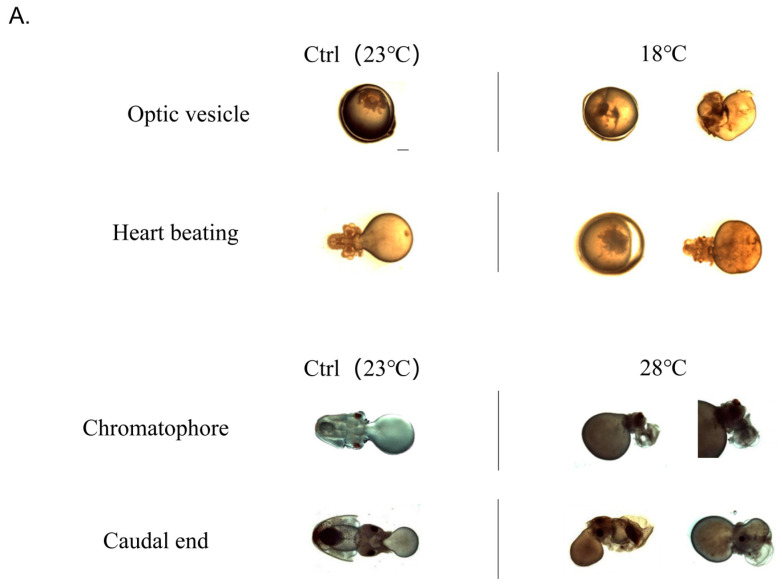
Embryonic malformation caused by low and high temperatures. (**A**) Morphological observation of embryos at significant malformation stages. All scale bars are 1 mm. (**B**) Histological observation of embryos at Caudal end stage. (**a**) Ctrl group (23 °C); (**b**) Low-temperature group (18 °C); (**c**) High-temperature group (28 °C). (MM: Mantle Muscle; VM: Visceral Mass; BR: Brain; YS: Yolk Sac; TA: Tentacular Arms). All scale bars are 200 μm.

**Figure 4 ijms-24-15365-f004:**
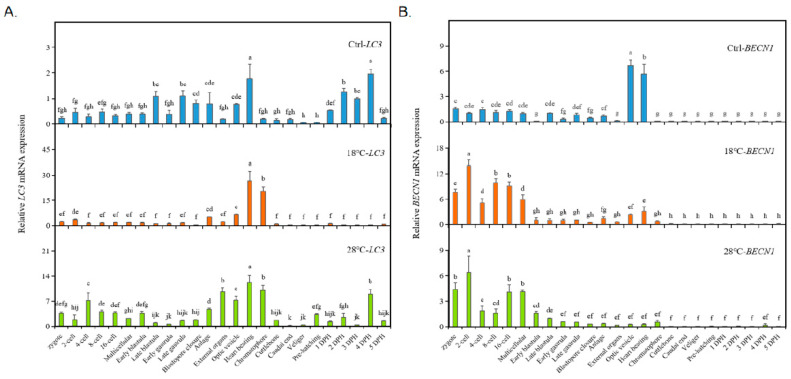
Expression of autophagy marker genes *LC3/BECN1* of the embryos. (**A**) *LC3.* (**B**) *BECN1*. The values are means ± SD (n = 12). Different letters denote significant differences between different development stages (*p* < 0.05).

**Figure 5 ijms-24-15365-f005:**
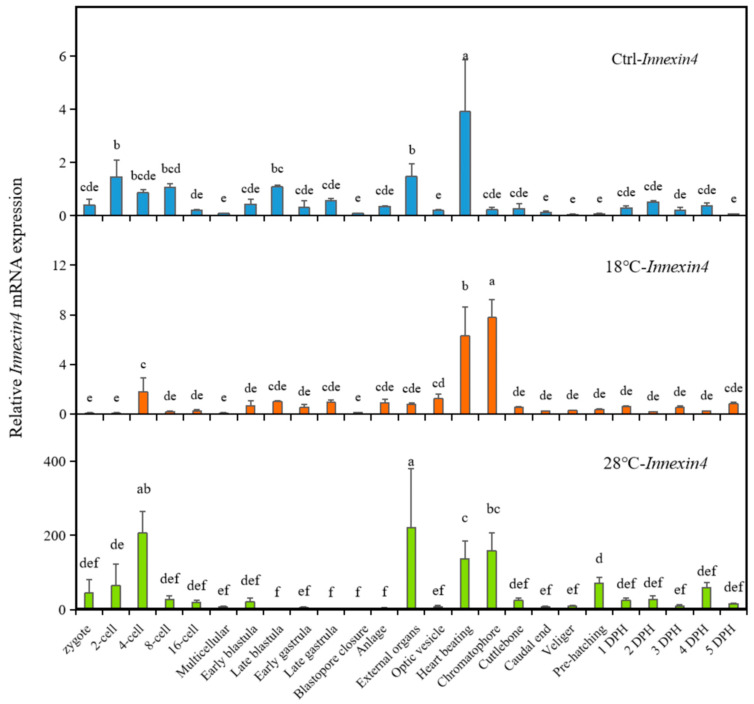
Expression of autophagy-related gene *Inx4* of the embryos. The values are means ± SD (n = 12). Different letters denote significant differences between different development stages (*p* < 0.05).

**Figure 6 ijms-24-15365-f006:**
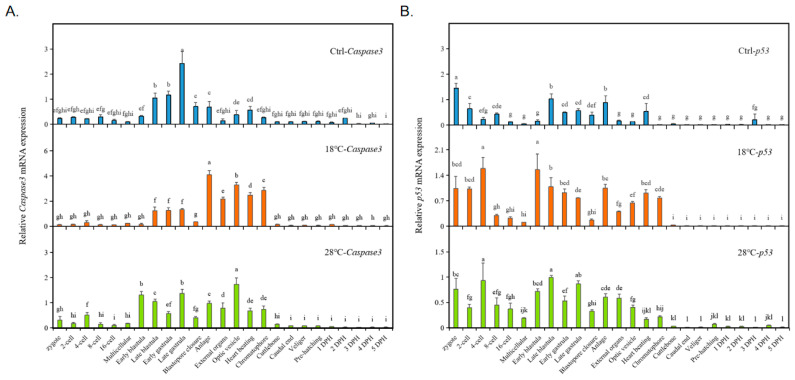
Expression of apoptosis marker genes *Cas3/p53* of the embryos. (**A**) *Cas3*; (**B**) *p53*. The values are means ± SD (n = 12). Different letters denote significant differences between different development stages (*p* < 0.05).

**Figure 7 ijms-24-15365-f007:**
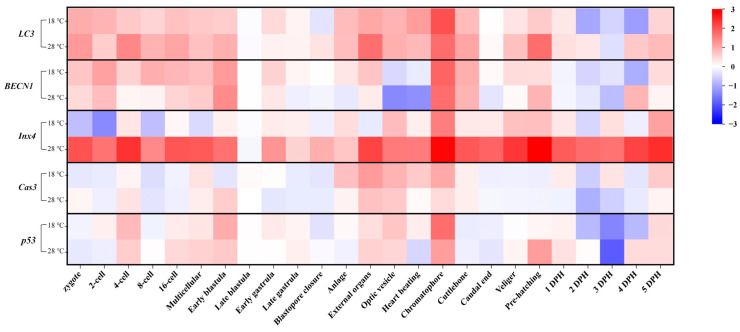
Heat map of the expressions elevated folds of *LC3*, *BECN1*, *Inx4*, *Cas3*, and *p53* at different developmental stages in the low- and high-temperature groups.

**Figure 8 ijms-24-15365-f008:**
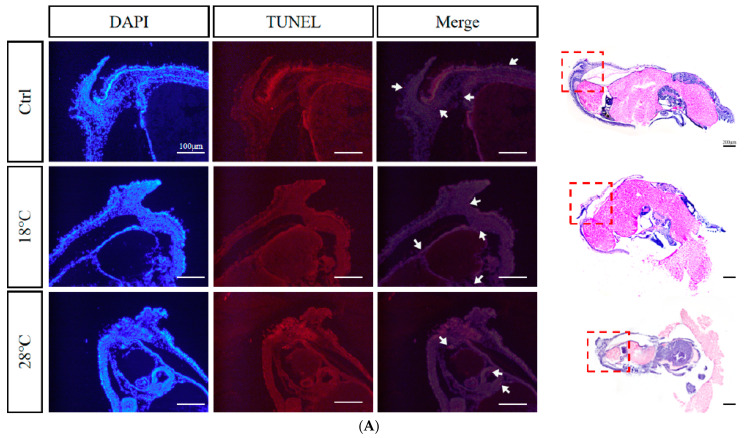
TUNEL apoptosis staining of malformed embryos of *S. japonica* at the Caudal end stage under different temperatures. The dotted red box circles the corresponding part. (**A**) Body; (**B**) Head; (**C**) Arms. All scale bars are 100 μm.

## Data Availability

The data presented in this study are available on request from the corresponding author. Informed consent was obtained from all subjects involved in the study.

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
