# Peer review of "The Effect of Temperature on the Embryo Development of Cephalopod Sepiella japonica Suggests Crosstalk between Autophagy and Apoptosis"

_ijms, 2023, doi:10.3390/ijms242015365_

Round 1

Reviewer 1 Report

Review ijms-2653066  

 The paper is valuable because includes valuable analysis between autophagy and apoptosis during embryonic development of Sepiella japonica under different temperatures. However, the paper includes well-planned analyses, there are some shortcomings in the experiment's and applied methods description.

It should include below information in the material and methods section:

1.     Please provide a description of the breeding procedure in Materials and methods section.

2.     How many mature specimens (females and males/pairs) were used in experimental reproduction? It should be included in the text of the manuscript.

3.     How many eggs were collected from each female?

4.     Please provide detailed information about the breeding and rearing facility. It was RAS or an open-water system? What was the cleaning procedure? Please provide more information of the chemical parameters of water (nitrite, nitrate, phosphorous) and photoperiod used during the experiment.

5.     Line 467 – salinity of 27 – please provide the unit of salinity.

6.     Line 468 – please provide the description of why 50% seawater renewal per day was conducted.

7.     Please provide the procedure of sample preparation to RNA isolation. Because the samples were different (from zygote to 5DPH) you need to prepare the samples in different ways. Please describe also the weight of samples used for RNA isolation. From how many samples in each experimental group the RNA isolation was conducted? Were the samples pooled?

8.     Please provide a description of the profile of qPCR analysis and reaction mix used in qPCR.

1.     In the Discussion section please provide the paragraph about the effect of low temperature on early development of sterlet published by Fopp-Bayat et al. 2022. Int. J. Mol. Sci. 2022, 23, 494. How Cold Shock Affects Ploidy Level and Early Ontogenetic Development of the Sterlet, A. ruthenus L. Int. J. Mol. Sci. 2022, 23, 494 https://doi.org/10.3390/ijms23010494

 I can accept this manuscript for publication after correcting the text during the above suggestions.

Reviewer 2 Report

Dear authors, 

Thank you for this very interesting paper. I only have some minor comments.

line 18, I suggest to delete "enigmatic creatures" 

line 42, are there natural waters of this temperature where S. japonicica will develop?

line 110. why are these creatures so mysterious? 

Figure 1/2/3 have a somewhat poor quality. Could this be adjusted? It is not clear to me what the letters below the different developmental stages mean. Please indicate this more clearly.

Line 122. please write DPH full out for the first time.  

figure 3B. maybe it would benefit to explain a bit more about the histological differences between the three groups. although you do, this is quit briefly described. For people without too much knowledge in the histology of cuttlefish this would be nice. In addition, the histological section of the HTG seems a bit macerated or broken down and if so could this influence morphology such as the ectropic mantle. moreover, was the abnormal curvature in the LTG also observed before fixation? In other words: how do you know that these are not artefacts from the histological process.  

line 184. mantle was distinctly different? This is somewhat vague. please explain in more depth.

Figure 4/5/6. again the letters are not clear for me what they denote. Could this be explained or depicted so that it becomes clear. 

Figure 8, very instructive and beautiful images.

Line 113. As you state that this paper holds significant scientific and industrial application value it would be nice to highlight this importance in the discussion which is now purely scientific with its developmental outcomes. 

I think this paper is well written and gives a nice overview of what happens to both developmental stages, autophagy and apoptosis in different experimental settings. As said it would be nice to include a bit more about its implication in a more broader sense.
